# Psychological State and Exam Performance among Paramedics’ Students in Geneva during the COVID-19 Pandemic: A Mixed Methods Study

**DOI:** 10.3390/ijerph20043736

**Published:** 2023-02-20

**Authors:** Florian Ozainne, Lou Rauss, Loric Stuby

**Affiliations:** 1École Supérieure de Soins Ambulanciers, College of Higher Education in Prehospital Care, CH-1231 Conches, Switzerland; 2Genève TEAM Ambulances, Emergency Medical Services, CH-1201 Geneva, Switzerland

**Keywords:** paramedic student, psychological distress, pandemic, COVID-19, performance, distance learning, stress, anxiety, mental health disorders, education

## Abstract

The COVID-19 pandemic forced higher education institutions in Switzerland to move to distance learning, with certain limitations such as “Zoom fatigue” and a lack of interaction with peers and teachers. This has also impacted the development of interprofessional skills and key concepts such as professional acknowledgement, cooperation, and communication skills. This study was conducted using mixed methods, including performance assessment regarding examination notes, the 12-item General Health Questionnaire, and semi-structured interviews, to assess the impact of the pandemic on the performance of paramedic students and their psychological state. The results of the semi-structured interviews provided insight into the impact of the pandemic. The period of the COVID-19 pandemic appears to have had an impact on the psychological state of the paramedic students, most of whom were considered to be either at risk or in psychological distress. There may have been an effect on their theoretical knowledge performance, with pre-pandemic promotions performing better than pandemic promotions.

## 1. Introduction

### 1.1. Background and Importance

In March 2020, the World Health Organization (WHO) announced that the coronavirus disease 2019 (COVID-19) outbreak could be characterized as a pandemic. In Switzerland, on March 13, 2020, the Federal Council declared an extraordinary situation and introduced more stringent measures, such as no complete lockdown [1]. For higher education institutions, this resulted in switching to distance courses and the cancellation of specific professional teaching methods such as learning technical skills, simulation [2], and problem-based learning in face-to-face small groups [3]. Indeed, the rationale for this study was to assess how distance learning due to COVID-19 impacts knowledge, practical skills, and psychological state. First, distance learning efficacy has some benefits and limitations, such as less travel time, flexibility, the ability for students to learn at their own pace, and opportunities for students to anonymously ask and answer questions, which could potentially encourage further engagement from those who would not otherwise participate in a live lecture due to the less intimidating environment online. The limitations frequently reported are family distractions, problems with internet connection, timing of tutorials, anxiety, and lack of space [4]. During the all-online course, videoconferencing seems exhausting for students. This fact is now well known and identified by the term “Zoom fatigue.” The components of this fatigue are interpersonal distance pressure (elevator syndrome), eye gaze at a close distance, cognitive load to produce and interpret nonverbal cues, reduced mobility, and mirror anxiety [5]. Women seem to be more affected by the mirror effect, an important component of this fatigue [6]. Fatigue and lack of sleep are factors that hinder learning.

The beneficial effects of the classroom courses are multiple. Positive peer influence, classroom cohesion, and student–teacher relationships are described as conditions for successful learning [7]. Empathy, warmth, sincerity, and genuineness are components of relationships that probably cannot be the same in videoconferencing. However, it is not clear to what extent distance affects psychological relationships. For example, telepsychiatry seems equivalent to face-to-face consultation [8].

For students and teachers, the perceived impact of COVID-19 on degree completion, perceived degree value, and future job prospects was a concern. In a previous study, medical students under lockdown reported feeling less prepared and felt that online teaching was less effective than face-to-face teaching [4]. Moreover, reading every day in the media that COVID-19 promotions have an uncertain future, that their competences may not be at the same level could impact belief in their self-efficacy [7]. In Hattie’s meta-analysis [7], collective teacher efficacy (the level of confidence teachers have in their ability to guide students to success) and estimates of achievement (the teacher’s belief about the level a student is able to achieve based on past experiences reflecting the accuracy of a teacher’s knowledge of their students, and not “teacher expectations”) could have a negative influence and produce the opposite of Pygmalion effect known as the Golem effect in this context [9].

During the curriculum, particular emphasis is normally placed on the development of interprofessional skills, as recommended by the Swiss Academy of Medical Sciences [10]. Interprofessionality is defined by the WHO as the learning and activity that occur when specialists from two or more professions work together. They learn from each other in the sense of effective collaboration that improves health outcomes [11]. In the prehospital setting, paramedics collaborate with a lot of different professionals, such as firefighters, police officers, nurses (e.g., hospitals, home care, medical-social institutions), medical radiology technicians, and emergency physicians. Developing interprofessional competence during initial training is a key component of safety in care. During the outbreak, all interprofessional learning activities were stopped to limit the spread of COVID-19, impacting the development of key concepts such as professional acknowledgement, reciprocity, and respect, cooperation, and communication skills, as well as turf protection, and workplace culture [12].

The cancellation of a portion of hospital and pre-hospital internships also resulted in less exposure for students to clinical practice until the end of April 2020. In Swiss universities under lockdown, isolation in social networks, a lack of interaction and emotional support, and physical isolation were associated with negative mental health issues such as stress, anxiety, loneliness, and depressive symptoms. Depression significantly prejudices learning, especially for women and those without adequate social support [13,14].

The term “promotion,” which will be used widely through the manuscript, is used to describe a timespan corresponding to a year of study, e.g., the 2019 promotion corresponds to the class of second-year students passing their paediatric exam in 2019.

A part of the students are commonly working as auxiliary staff in emergency medical services (EMS) from the second year of training, which can accumulate negative effects on learning, and expose them to aggravated psychological pressure, and even mental illness during the COVID-19 outbreak [15].

In Switzerland, there has been no study of the COVID-19 pandemic’s impact in the specific setting of vocational education. The purpose of this study was to identify the effect of this period on learning and mental health in paramedic students.

### 1.2. Objectives

The main objective was to assess the impact of the lockdown and the learning adaptations during the COVID-19 pandemic on the promotions of paramedic students compared to pre-pandemic promotions regarding a specific exam performance.

The secondary objective was to screen students from pandemic promotions for psychological disorders using the 12-item General Health Questionnaire (GHQ-12) and semi-structured interviews.

## 2. Materials and Methods

### 2.1. Study Design

The study was based on mixed methods. Three methods were used in this study: firstly, part one of the quantitative approach used the examination notes of the six last promotions to determine the students’ performance. Then, the GHQ-12 questionnaire was used to assess the psychological distress of the pandemic promotions. Finally, semi-structured interviews were conducted among some students from the pandemic promotions to explore the underlying reasons for the quantitative results.

This study adheres to the STROBE (Strengthening the Reporting of Observational Studies in Epidemiology) statement [16] (Appendix A). Qualitative data results are presented according to the COREQ (Consolidated Criteria for Reporting Qualitative Research) checklist [17] (Appendix A).

### 2.2. Outcomes

The primary outcome was the score obtained in a specific examination, i.e., the paediatric one. The exam questionnaires are regularly updated and renewed, making it impossible to compare between promotions. However, the paediatric exam has hardly changed over the last six years, justifying its use. Although they have several exams during their curriculum, it was therefore decided to only assess the 2nd year paediatric examination score. In Switzerland, there are several ways to calculate examination scores [18], the most common being the Swiss federal scale, i.e., applying the formula: s = 5(p/t) + 1, where s = examination score ranging from 1.0 to 6.0, p = amount of points obtained by the learner, and t = the total amount of points. The Geneva’s College of Higher Education in Prehospital Care, is applying this scale when the total of points is up to 150. When the total of points exceeds 150, the applied formula is the following: s = 5(p/t). This latter formula was the one used to calculate the scores used in the current study, with the specific paediatric exam containing 178 possible points.

The secondary outcomes were the scores of the GHQ-12 questionnaire, and the answers gathered through the semi-structured interviews.

### 2.3. Ethics

A clarification of responsibility was submitted to the local ethics committee (“Commission Cantonale d’Éthique de la Recherche sur l’être humain de Genève”). This study was considered to fall outside the scope of the Swiss legislation regulating research on human subjects, so the need for local ethics committee approval was waived (Req-2022-00361).

### 2.4. Settings, and Time Frame

For the primary outcome analysis, the setting was constituted of 118 students over six promotions (among which 4 promotions passed the examination before the pandemic, i.e., from 2016 to 2019, and 2 promotions during the pandemic, i.e., in 2020, and 2021). The choice of the number of promotions was based on the version of the examination, which has changed little over this period.

For the GHQ-12, the study population consisted of 64 students in three grades (first, second, and third year) studying at the College of Higher Education in Prehospital Care in May 2021. They had 4 days to complete the questionnaire (between the 17 May 2021 and the 20 May 2021).

For the qualitative part of the study, a convenience sample of six students was selected on a voluntary basis following these criteria: two per grade (during the first epidemic outbreak and the lockdown from 13 March to 27 April 2020), one of each main gender. The interviews were conducted between February 2022 and March 2022.

No participant received any compensation.

### 2.5. Sample Size Calculation

One hundred and eight participants were required to have a 90% chance of detecting, at the 5% level, a significant change in the examination note from 0.25 points in the pre-pandemic group compared to the pandemic group (e.g., an examination score change from 3.5 to 3.75 or from 4.5 to 4.25). A timeframe of six promotions (i.e., years) was selected because this was the period during which the examination remained the most unchanged. This timeframe represented 118 students’ examination performance. Given our sample size calculation, an adequate size could be achieved using this data. For the GHQ-12 outcome and to be as exhaustive as possible, the questionnaire was sent to a convenience sample containing all students from the pandemic promotions. No data exists on pre-pandemic promotions, and we did not want to recall them to limit memory biases.

### 2.6. 12-Item General Health Questionnaire

The General Health Questionnaire (GHQ) was developed in 1970 by David Goldberg and Paul Williams [19]. It is used to screen for minor psychiatric or psychological disorders in the general population or in non-psychotic patients [20]. The original language of this questionnaire is English, and it has been translated into several languages, including French [21]. Several long versions of the GHQ exist, ranging from 28 to 60 questions. As the name suggests, the GHQ-12 consists of 12 questions. It is designed to screen for psychological distress by looking for symptoms among the following 4 items: depression, anxiety, social maladjustment, and somatic complaints.

Respondents to this questionnaire should consider their current condition and the weeks leading up to it. There are 4 response modalities for the different items (4-item Likert scale), e.g., “Better than usual,” “As usual,” “Less well than usual,” and “Much less well than usual.” There are several ways to account for the results. Before data collection, we decided to dichotomize the results, with the two first modalities corresponding to the absence of a problem (rated as 0) and the last two responses corresponding to the existence of suffering (rated as 1) [22,23]. With this bimodal scoring system, the total score is twelve.

The prespecified thresholds were less than 2—mentally healthy, 2—at risk, and over 2—in psychological distress. The optimal threshold has been reported to range from 1/2 to 6/7, with the most common threshold being 2/3. If the goal of screening for psychiatric disorders in primary care is to help bridge treatment gaps, the recommended threshold is 1/2 point, where a score of 2 or above is ‘positive’ for those at risk of psychiatric disorders [23,24].

### 2.7. Semi-Structured Interviews

Semi-structured interviews were conducted by Lou Rauss (L.R.), a third-year female paramedic student without prior specific experience, for the purpose of her diploma thesis. She was studying at the College of Higher Education in Prehospital Care in Geneva, so she was also impacted by the pandemic. Prior to the study, a relationship was established with the participants, as they all attended the same college as her, particularly the third-year students who were in the same class. The participants were informed about the reasons underlying the study, and a written consent was gathered retrospectively. Content analysis was the methodological orientation stated to underpin the study. Saturation could not be determined as no exploratory factor analysis of the interview guide was conducted.

A convenience sample of two students (a male and a female) was purposely selected by promotion for their willingness to take part in the study, their ability to express their feelings clearly, and to take an inside look. These students have all experienced semi-lockdown. The interviews were carried out twice face-to-face and the others by video conference. Notes were written during the interviews, lasting about 20 to 25 min, which were also audio recorded to allow further retranscription. No one, apart from the interviewer and the participant, attended the discussion. An interview guide (Appendix A), created by L.R. and tested by Florian Ozainne (F.O.), consisted of several parts that focused mainly on the semi-lockdown period. Themes were identified in advance by L.R. and F.O. based on the existing literature.

The semi-structured interviews were based on a thematic analysis, according to Dejonckheere, M., and Vaughn, L.M. [25]. The first part was about experiencing the lockdown and the emotions felt during this period (such as joy, sadness, disgust, fear, and anger). The second part explored the social support dimension (loneliness, family, and friend support). The third part was about experiencing (theoretical) distance learning. The next part explored students’ and teachers’ social support. The fifth part focused on practical training (such as internships and simulated practice at school). To conclude, we searched for the technique used for distance learning.

Once the interviews were done, L.R. transcribed them and sent them to F.O. and Loric Stuby for interpretation. The main themes were identified individually before the interpretation was brought together. Transcripts were not returned to participants for comment and/or correction. They were either not asked to provide feedback on the findings.

### 2.8. Statistical Analysis

The continuous variables were described using either the mean (95% confidence interval or SD) and/or median (quartiles +/− range) and tested with the student’s *t*-test or the Mann–Whitney U test according to normality. Proportions were tested using Fisher’s exact test. Distribution was assessed graphically, and, in doubt, using the Shapiro-Wilk test. The examination note was treated as a continuous variable (unrounded). Missing data was treated as such. An univariable linear regression model was generated to assess the prediction of the examination scores based on the promotion. The objective was to compare the promotions between them, looking for “a promotion effect” that could affect the overall result. The quality of the model was assessed by checking the assumptions. Considering the rule proposed by Stevens [26], i.e., 15 participants per predictor, with 6 predictors (2016, 2017, 2018, 2019, 2020, and 2021 promotions), the sample size should be 90 subjects. When using the rule proposed by Tabachnik and Fidell [27], i.e., N > 50 + 8*number of predictors, the sample size should be 98 subjects. Accordingly, the sample size of 118 subjects seems adequate to apply this statistical method. Two sensitivity analyses were then performed based on statistically significant differences between the groups, i.e., by first including only students originating from the Canton of Geneva, then by excluding the 2019′s promotion. Proportions were tested using Fisher’s exact test. The reliability of the GHQ-12 was measured using Cronbach’s alpha. A two-sided p-value lower than 0.05 was considered significant. All statistical analyses were performed using Stata V15.1 (StataCorp. 2017. Stata Statistical Software: Release 15. College Station, TX, USA, StataCorp LLC).

## 3. Results

### 3.1. Examination Score

The characteristics of the 118 students (100%, 118/118) included in the primary outcome analysis are displayed in Table 1.

The mean examination notes were different among groups [mean (95%CI), pre-pandemic 4.99 (4.91 to 5.07) versus pandemic 4.75 (4.63 to 4.88)]. (Figure 1).

As the canton of origin differs among groups (Table 1), a sensitivity analysis was conducted by analysing only the subjects issued from the canton of Geneva. Among Geneva’s students only, the difference remained to about the same extent (pre-pandemic 5.06 (4.95 to 5.17) versus pandemic 4.73 (4.57 to 4.90)).

A simple linear regression was generated to predict the examination scores based on the promotion. A significant regression equation was found (F(5, 112) = 5.13, *p* < 0.001), with an *R*^2^ of 0.19. The results by promotion are displayed in Table 2.

As the 2019 promotion was different from the others (and could influence the overall outcome with a greater promotion effect than the pandemic effect), a second sensitivity analysis was performed by reconducting the primary analysis with the exclusion of this promotion. The difference remained similar between groups after the exclusion of 2019′s promotion (pre-pandemic 4.92 (4.83 to 5.01) versus pandemic 4.75 (4.63 to 4.88), *p* = 0.03).

### 3.2. GHQ-12

The GHQ-12 was sent to 64 students, of whom 52 (81.3%) answered. Their characteristics are displayed in Table 3. The questionnaire consisted of 12 items, and the value for Cronbach’s alpha was α = 0.85.

The median (Q1; Q3), [min; max] of the GHQ-12 was 2 (1; 6), [0; 11]. The mean (SD) was 3.71 (3.12). The distribution is shown in Figure 2.

Using the prespecified thresholds (<2—mentally healthy, 2—at risk, and >2—in psychological distress, the students were categorised as follows: 16 (30.8%) were healthy, 11 (21.2%) were at risk, and 25 (48.1%) were in psychological distress.

### 3.3. Semi-Structured Interviews

Six students were selected to participate in the semi-structured interviews. Their characteristics are reported in Table 4.

#### 3.3.1. The Experience of Semi-Lockdown

We noted that all the students interviewed experienced the semi-lockdown more or less well. In fact, students who were professionally involved in a service during this period experienced it better. They could *“feel useful.”* A student reported that he *“Could be useful and support the EMS,”* another explained that he *“Could still be active, [...] I had a chance to work in an EMS because our internships were all cancelled.”*

#### 3.3.2. Stress and Emotions

The students’ testimonies reported very different feelings. Stress was present in all the students, either because of concern about the disease or about the future of the curriculum. For one of them, who worked in an EMS, the stress was caused by the fear of infecting his loved ones. He explains that *“It was a very stressful time because my mom is immunosuppressed. I was potentially a vector. I had to move and live with my best friend.”* Fear was present in some of them, linked to their relatives, and in others, to a lack of knowledge and information about the virus. Anxiety was a feeling expressed by all students. A student reported that “*you are permanently anxious and then you can’t get into your classes or anything.”* One student said she was *“Worried about becoming incompetent*” in her future occupation as a paramedic. Contrary to expectations sadness was not noted. Anger and joy were expressed by some students. Anger was expressed by doubting the college. *“It was when we were told by email that they had announced to the services that they were closing the internships. I think that was the first time I started to doubt the college.”*

#### 3.3.3. Somatic Manifestations

Some students reported somatic manifestations of their anxiety, and stress. One student said: *“You close yourself up, tummy ache [...]”*; and another said: *“I was moving less, I gained quite a bit of weight and that stressed me a lot too [...]”*.

#### 3.3.4. Social Dimension, Isolation, Loneliness

Most students felt lonely during this period. First-year students who have not been assigned to a workplace feel lonelier. According to one student, *“you isolate yourself, you don’t want to see people anymore”* and *“you get away from everyone.”* Despite this loneliness, they all lived in shared apartments or with their parents.

They expressed a kind of after-effect of semi-lockdown like a lack of motivation to go out. This was illustrated by the words of the student, who *“lost the pleasure to go out and see people, drink beers.”* This was despite a large network of friends for most students. One student said, *“You get away from everyone a bit.”*

#### 3.3.5. Distance Learning

In most interviews, students said that distance learning was difficult. The presence of difficulties in the learning of some students did not favour a good adherence to distance learning, especially videoconferences.

Nearly all students described home as a quiet place. However, home was not a good place to learn since there was a lot of entertainment: “*Very hard to follow the lessons, to concentrate [...]. Always distracted at home.”*; *“Because of the video conferencing, it was very difficult to concentrate.”*; *“Very hard [...]. And in videoconferencing even more so, being all alone in my room in front of my screen, I could not concentrate at all.”*; “*I have too many distractions at home.”*; “*It’s hard to stay focused, your attention quickly wanders.”*; *“You can’t ask all the questions you want.”*; “*As long as you are at home, it is not conducive to learning. I had a quiet environment but even so, I had plenty of distractions.”*

The lack of interaction and *“human presence”* was expressed by all students during distance learning courses as well as in a working environment: “*I missed the interactions the most, because they are very difficult in video conferencing.”*; *“And also the contact with the other students, to see them at breaks, to discuss with them.”*; “*Because of the lack of interaction, I had less concentration, it was harder to follow a class and pay attention.”*; *“Usually, when we were in class and I did not understand, I often had the opportunity to ask the question in a low voice to a classmate who answered me in two words, and allowed me to stay focused.”*; *“What I missed the most was the exchange, it’s still different from face-to-face, the exchange with your classmates, where you can ask questions.”*

A student also mentioned logistical issues: “*It was very complicated with connection problems, computer problems, sound problems. [...] with much less interaction between people*.*”*

Furthermore, half of the students were able to find, through their families, help with learning. However, all the students were able to find help from a classmate. “*There was quite a bit of support.”* Only one of the students appreciated the learning by videoconference; he had previous experience with distance learning. All the interviewed students appreciated the e-learning and the recorded lectures, which allowed them to work at their own pace. One student explained that “*the only good thing is the recorded lectures that you can listen to again.*”

#### 3.3.6. Internship, Practice, Skills

All students were impacted by the reduction in the number of training hours during the semi-lockdown period. Only the first-year students were able to do their prehospital internship. The internships of the second- and third-year students were cancelled, and many of the practical exercises were removed. The latter did not have their internship in specific fields such as anaesthesia or paediatrics. The consequences of this absence of about 2 months were felt. One student illustrated this with these words: “*Personally, paediatrics remains a stress that I do not have in the other branches. The only thing I missed was paediatrics, and I feel like that’s where it comes from, even though paediatrics is still something that is very stressful for everyone.”*.

Students were unanimous in their view that the pandemic has had an impact on the development of their practical skills. However, they did not notice a difference in their new occupation. Third-year students described their lack of skills in the areas where the internship was cancelled: “*I am comfortable with COVID-19 management but I am not comfortable with management of complications of childbirth, or major events*.” The development of certain specific skills also seems to have been impacted: *“I think if I had been able to do my anaesthesiology internship I would be even more comfortable than I am now with ventilation and medication.”* For the third-year students, the return to the internship never took place, and they immediately attended their final exams.

#### 3.3.7. Motivation and Commitment to the Curriculum

Motivation was a challenge for all students. For some, videoconferencing courses required more work*: “When it was a whole week of videoconferencing, I was demotivated”; “It takes twice as much work because you have to redo all the courses because it is very difficult to follow.”*

For others, the fact of being alone in front of a screen and not being able to interact was demotivating: *“It’s not very motivating to get up and listen to someone you don’t interact with…”*

One student also mentioned the absence of diversity in prehospital care: *“Motivation has changed, because non-diversified interventions during two months of internship is complicated.”*

However, none of them had any doubts about their motivation to continue the curriculum.

#### 3.3.8. Consideration of the Diploma

A student has questioned herself about her level: *“I was very worried that we would graduate with less competence as the others.”*

The title of paramedic for students who graduated in July 2021 has been criticized by some graduated paramedics due to their lack of practicum as well as the way they passed their final exam: “*Many people in the professional community criticized the management of the exams, and said that the exam was poorly administered by the college and that many students should not have passed because they did not have the skills.”* However, two years after graduation, those who were third-year students during the first lockdown did not feel that the pandemic had changed the value of their degree.

#### 3.3.9. Coping Strategies

The coping strategies applied by the students were all different, they did activities to entertain themselves: *“Puzzles, embroidery, knitting, video games, activities that kept my mind busy.”; “I tried to do sports at home and keep a balance between physical activity, and work for college.”*

They also tried to socialize themselves: *“I moved in with my best friend, I think it helped me keep a balance.”*; or to stay informed: *“I got informed by reading studies, and by being confronted with the disease.”; “I have made inquiries about the disease, and what was going on. I think that was the best way to fight the fear of the unknown.”*

One student mentioned the need to have a frame: *“We had to set ourselves deadlines [...]. I made a schedule for the rest of the lockdown [...].”* For some, coping strategies in themselves seemed difficult to live: *“You run away to something else, you do activities where you escape. Sports, series, or stuff like that. It is not about reducing stress but forgetting about it.”*

#### 3.3.10. Commitment to an Emergency Medical Services

Most students reported working in an EMS. Their testimonies highlight the positive aspect of being active in the field during this period: *“I could be useful and support the EMS and when we were in confinement we could work a lot, and that’s cool”*; *“As soon as the lockdown started, I very quickly started working in EMS so I did not feel isolated at all at any point. I could still be active. I had a chance to work in an ambulance because our internships were all cancelled.” “I was working in an EMS, I always felt invested”. “Working in an ambulance service saved my degree.”* In these students, two reported to being stressed by working in EMS: *“It was a very stressful time because my mom is immunosuppressed. I was potentially a vector. I had to move and live with my best friend.” “I was afraid at the beginning of the pandemic. But once we had more explanation about the virus, that I could be confronted with the disease, with the protective gear...it was okay... it reduced my fear a lot...”*

## 4. Discussion

The COVID-19 outbreak impacts scholar performance with a mean score (95% CI), pre-pandemic 4.99 out of 6 points (4.91 to 5.07) versus pandemic 4.75 out of 6 points (4.63 to 4.88)] in these specific exams. This is relevant because it represents one grade difference (while 4.99 could correspond to a B-/C+ grade, 4.75 corresponds to a C) [18]. 

The effect of the COVID-19 pandemic on student performance could be related to many factors.

The shift to online distance learning in vocational education could be beneficial. Typically, the curriculum includes more than 50% hands-on skills labs and simulations. However, due to the outbreak, emergency adaptations from blended learning to 100% online courses must have been made. Effective online courses need to be adjusted and pedagogical concepts should be designed and developed explicitly for e-learning [28]. 

Internet connection problems are a common barrier to effective online courses [4]. As highlighted by a student, “*It was very complicated with connection problems, computer problems, sound problems,”* demonstrating the difficulties of implementing a 100% online program. 

The motivational aspect has an impact, with boredom having a negative impact and motivation having a positive impact [29]. Two students reported: *“When it was a whole week of videoconferencing, I was demotivated,”* and *“It’s not very motivating to get up and listen to someone you don’t interact with.”* Low motivation seems to have occurred in at least these two students. 

Family distractions and concentrating difficulties are frequent barriers to effective learning [4]. In our sample, most students reported lots of distractions and difficulties concentrating at home. For example, they reported sentences such as *“Always distracted at home.”*; *“When video conferencing, it was very difficult to concentrate.”*

The screen time was increased, both due to distance learning and for entertainment [30], as emphasized by the student, who reported, *“you escape into something else; you do activities where you escape. […] watch series, things like that.”* This could impact learning, a scholar’s results, and their psychological state in different ways. Higher levels of screen time were associated with a variety of health harms for children and young people, with evidence for adiposity/overweight, an unhealthy diet, depressive symptoms, and quality of life [31,32]. By the way, one of the students in the sample reported that he/she was less physically active and had gained weight. Evening screen use could also impair the quality of sleep [33]. Some authors documented an association between screen time and self-esteem, poorer mental health, and more severe depressive and anxiety symptoms [34,35]. 

About self-esteem and self-efficacy, a student reported this feeling: “*I was very worried that we would graduate with less competence as the others.”* Belief in self-efficacy is an important predictor of successful learning too [29,36]. This was not formally evaluated in our study, but the same feeling was experienced by the teachers about how the “COVID promotions” were impacted once in the field. This is another possible opposite to the Pygmalion effect, named the Golem effect [9].

The hands-on workshops and simulations were discontinued. These workshops are, of course, important for the development of practical skills, but not only. They are always followed by a debriefing where the practical aspect is put into perspective with the theoretical learning, allowing a good anchoring of the knowledge. Blended learning with face-to-face courses, especially in vocational training, makes sense. Psychomotor skills cannot be developed through video conferencing. For example, Currat et al. found that adding face-to-face training to an e-learning module increased proficiency and enhanced skill retention in student paramedics [37,38].

All interprofessional learning activities were stopped in the first wave of COVID-19. To illustrate that, a student reported that he *“was pissed off”* when the school announced the cancellation of the 24-h high-fidelity simulation with nurses, emergency physicians, firefighters, policemen, and policewomen. It takes time to understand others’ priorities and problems. This impacts the development of key concepts such as professional acknowledgement, reciprocity and respect, cooperation, communication skills, working on protecting turf, and workplace culture [12]. Interprofessional learning is a cornerstone of teamwork and patient safety. The same holds true for the development of interpersonal skills such as empathy. We did not specifically measure this effect. There are interprofessional competency assessment tools to measure this parameter. This could be an area for future research. 

In the general population, the GHQ-12 mean score varies from 1.63 points (SD 1.98) in Korea [39] to 5.9 across Europe in primary care clinics [40]. In healthcare professions, for Portuguese medical students, the mean was 5.77 [24] and was 4.50 (SD 2.89) for nursing students during their final study year [41]. Our students were therefore in the middle of the range.

Closer to our population, a study among French-speaking Swiss paramedics in 2011 [42], using GHQ-12 scores with cut-off points of 2/3, showed that the percentages of potential cases with poor mental health were 20%. It is less than our sample, where nearly half of the students suffered psychologically during this period.

A part of the GHQ-12 score was related to depressive symptoms, and anxiety. In semi-structured interviews, a large portion of students used words such as “anxiety, fear.” One student reported fear of infecting his relatives. Similar findings were noted in a descriptive study of baccalaureate nursing students, where 84% reported that it was a major cause of fear, followed by the fear of contracting COVID-19 [43]. In December 2022, the federal office of public health published a report showing a 26% increase in hospitalizations for suicide attempts between 2020 and 2021 in young people aged 10 to 24 years old. Mental disorders were for the first time the leading cause of hospitalizations among this population, ahead of injuries [44]. Severe depressive symptoms were frequent in the 14–24-years-old group, with an increasing proportion over time [45]. The mean age of students in our study was 24.8 years old, showing that they were encompassed by the conclusions of these reports. It can explain in part the results of the GHQ-12, and the scores of the exams. Indeed, psychological distress, anxiety, and fear are well known to make learning and memorization more difficult. In Hattie’s work about visible learning in high school [7], depressive symptoms, and anxiety are major negative factors, and that could explain partly the result of the theoretical knowledge exam observed in our study.

Fatigue, and lack of sleep are not good conditions for learning. As described above, major parts of the courses were conducted via videoconferencing. Clearly, this allowed the curriculum to be continued, but the “Zoom fatigue” effect could also be a barrier to learning, as mentioned by Hattie about a lack of sleep [29]. 

In the semi-structured interviews, students involved in EMS have more positive discourses on the experience of COVID-19 outbreak. In comparison, a study among emergency medical workers in Spain showed that the COVID-19 pandemic may have been a traumatic event. Indeed, one-third of them reported a possible diagnosis of post-traumatic stress disorder in the emergency department [46]. The mean score in the GHQ-12 was 5.26 (SD 3.18), therefore higher than our result. We can assume that students accumulate negative effects and are exposed to aggravated psychological pressure and even mental illness during the COVID-19 outbreak by working in EMS [15,46]. No subgroup analysis of the GHQ-12 stratified on the working status (working or not) was done. However, we could hypothesize, based on the semi-structured interviews, that feeling useful, being in action, and helping EMS could probably be protective factors. It is illustrated by these statements: “[...] *could be useful and support the ambulance services and when we were in confinement we could work a lot, and that’s cool”*; *“As soon as the lockdown started, I very quickly started working in EMS so I didn’t feel isolated at all at any point. I could still be active. I had a chance to work in an ambulance because our internships were all cancelled.” “I was working in an EMS, I always felt invested.” “Working in an ambulance service saved my degree.”*

These testimonies seem to show that the perception of control over a stressful experience could determine its impact on the individual. Helplessness against adversities could be a factor in stress-related psychiatric illnesses, such as depression, anxiety, and post-traumatic stress disorder [47]. Conversely, feeling that challenging situations are manageable is associated with resilience and positive outcomes [48,49]. Levine highlights this in his book about healing psychological trauma. “If you can neither flee nor fight, you freeze your psychic life, but if you are helped to act physically or mentally, you can escape the trauma” [50].

Despite the wide psychological support offered (by the college, by the state of Geneva, by two associations, and through peer support among students and professionals), these results show that there is room for improvement, in particular in being more proactive to ensure the well-being of students.

The study has several limitations. The primary analysis was based on only one specific exam, thus not necessarily reflecting the actual global performance of the paramedic students. The choice of thresholds for the GHQ-12 analysis could be debated, but we reported the detailed results, thus allowing their interpretation with different thresholds. The samples were different for each analysis, preventing us from assessing the same participants using different approaches. It does, however, allow the depiction of an overview. The semi-structured interviews were conducted about 18 months after the lockdown, which leaves the results subject to recall bias. The sample size was extremely limited, thus restricting the generalizability. As with all qualitative studies, it is never possible to guarantee that another study sample would not have led to a different result. Nevertheless, the data collected provided rich material describing many different individual situations, and experiences.

## 5. Conclusions

The COVID-19 pandemic period appears to have had an impact on the psychological state of the paramedic students, with an effect on their theoretical knowledge performance.

The lessons learned from this pandemic should help train institutions to better anticipate this type of crisis. First, by reinforcing social and psychological support in cases of interruption of courses, by developing a reliable tool dedicated to distance learning and designed pedagogically for this purpose (e.g., e-learning with a pedagogical path, video, and feedback), and by training teachers to use this tool so that they do not simply put online contents usually taught face-to-face. In view of the decline in school results, tutoring should also be considered.

Students represent a resource that can be mobilized in a context of crisis, all the more because students who were actively involved on the front line seemed to experience the period better than those who stayed passively at home.

## Figures and Tables

**Figure 1 ijerph-20-03736-f001:**
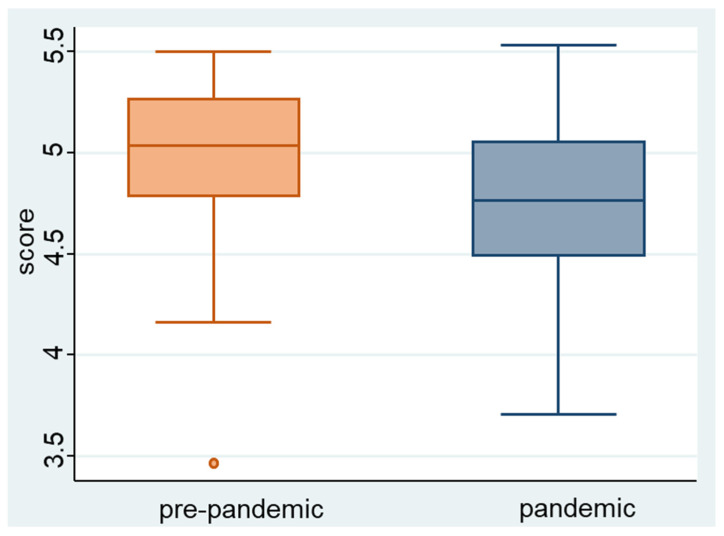
Examination score by period.

**Figure 2 ijerph-20-03736-f002:**
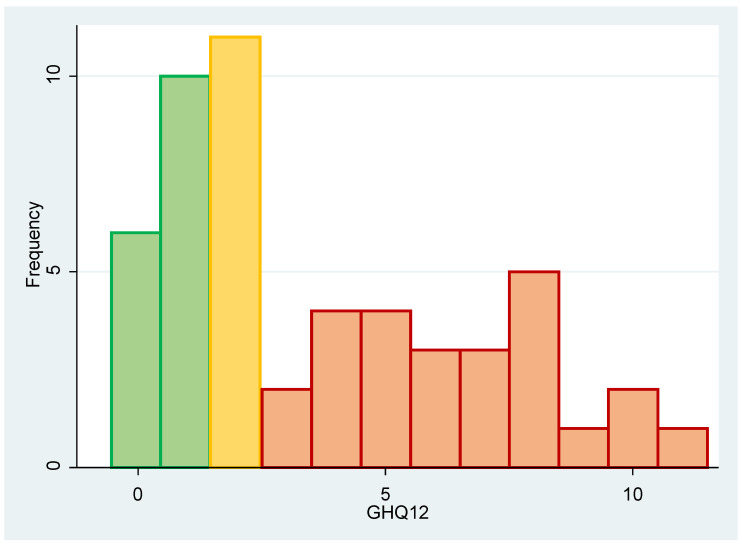
GHQ-12 results’ frequency distribution (with the colours representing the three categories: green = healthy, orange = at risk, red = in psychological distress).

**Table 1 ijerph-20-03736-t001:** Characteristics of students included in primary outcome analysis, by group.

Item	Pre-Pandemic (n = 75)	Pandemic (n = 43)	*p*-Value
Gender, n (%) -Female -Male -Other	33 (44.0) 42 (56.0) 0 (0.0)	24 (55.8) 19 (44.2) 0 (0.0)	0.25 *
Age, median (Q1; Q3)	24 (22; 27)	24 (23; 25)	0.61 ^†^
Canton of origin, n (%) -Fribourg -Geneva -Jura -Neuchâtel -Vaud -Valais - Other: France	7 (9.3) 25 (33.3) 3 (4.0) 0 (0.0) 27 (36.0) 10 (13.3) 3 (4.0)	2 (4.7) 26 (60.5) 1 (2.3) 2 (4.7) 8 (18.6) 3 (7.0) 1 (2.3)	0.04 *
Promotion, n (%) -2016 -2017 -2018 -2019 -2020 -2021	21 (28.0) 21 (28.0) 17 (22.7) 16 (21.3) N/A ^1^ N/A ^1^	N/A ^1^ N/A ^1^ N/A ^1^ N/A ^1^ 23 (53.5) 20 (46.5)	N/A ^1^

^1^ N/A: not applicable; * Fisher’s exact test; ^†^ Mann–Whitney U test.

**Table 2 ijerph-20-03736-t002:** Examination score by promotion, calculated by linear regression, and expressed as mean (95%CI).

Promotion	Coefficient (95% CI)	Score (95% CI)	*p*-Value ^1^
2016	Ref.	4.83 (4.71 to 4.95)	Ref.
2017	0.10 (−0.11 to 0.32)	4.93 (4.73 to 5.14)	0.35
2018	0.20 (−0.03 to 0.42)	5.03 (4.88 to 5.18)	0.09
2019	0.41 (0.18 to 0.65)	5.24 (5.14 to 5.35)	0.001
2020	−0.07 (−0.28 to 0.14)	4.76 (4.60 to 4.92)	0.50
2021	−0.08 (−0.30 to 0.14)	4.75 (4.54 to 4.96)	0.46

^1^ Obtained by univariable linear regression.

**Table 3 ijerph-20-03736-t003:** Characteristics of students who answered the GHQ-12.

Item	Result (n = 52)
Gender, n (%) -Female -Male -Other	27 (51.9) 23 (44.2) 2 (3.9)
Age, median (Q1; Q3)	24 (22; 27)
Canton of residence, n (%) -Fribourg -Geneva -Neuchâtel -Vaud -Valais -Missing	3 (5.8) 34 (65.4) 3 (5.8) 5 (9.6) 6 (11.5) 1 (1.9)
Grade, n (%) -First-year -Second-year -Third-year	19 (36.5) 18 (34.6) 15 (28.9)

**Table 4 ijerph-20-03736-t004:** Characteristics of students who took part in the interviews.

Item	Result (n = 6)
Gender, n (%) -Female -Male -Other	3 (50.0) 3 (50.0) 0 (0.0)
Age, median (Q1; Q3)	24 (24; 26)
Grade, n (%) -First-year -Second-year -Third-year	2 (33.3) 2 (33.3) 2 (33.3)

## Data Availability

The data presented in this study are available as Appendix A.

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
