# Peer review of "Psychological State and Exam Performance among Paramedics’ Students in Geneva during the COVID-19 Pandemic: A Mixed Methods Study"

_ijerph, 2023, doi:10.3390/ijerph20043736_

Round 1

Reviewer 1 Report

Thank you for the opportunity to review for the International journal of environmental research and public health. The manuscript is entitled titled ‘Psychological distress and exam performance among paramedics’ students in Geneva during the COVID-19 pandemic: a mixed methods study’. While I find this article interesting as it tries to connect examination grades, stress levels, and personal feelings during pandemic circumstances, gained through the interviews, I am not convinced that the arguments and methods justify the conclusion. My detailed comments are provided below.

Introduction

The main objective of this study is ‘to assess the impact of the lockdown, and the learning adaptations during the COVID-19 pandemic on promotions of paramedics’ students compared to pre-pandemic promotions regarding a specific exam performance.’ However, the idea behind promotions is not described in the introduction or in the remaining text. The authors used this word with different meanings. For instance, they mentioned the details of work experience in Emergency Medical Centers which can be related to promotions in organizations. Later, in Table 2, promotions were presented as calendar years. As a result, it is not clear enough either promotion is considered in the sense of career development or educational progress or even a simple timespan.

The overall impression is that the introductory part is quite vague and not perfectly related to the main objective. More contextual details, such as the description of promotion, can be valuable for this section.

Materials and Methods

Line 143 ‘Given our sample calculation, the adequate size could be achieved using these data.’ – please explain how this calculation was performed, and what references were used in which such a sample size was considered as adequate for further analysis.

Line 147 ‘For the GHQ-12 outcome, a convenience sample was used’ - How was the adequate sample size reached?

Line 151 ‘The General Health Questionnaire (GHQ)’ - It would be more convenient for readers if you provided this explanation of the abbreviation together with the first mention of this questionnaire.

The authors should provide the reliability of the GHQ-12 questionnaire for the current sample (Cronbach’s alpha coefficients, composite reliability (CR), or the like).

The paragraph (lines 166-172) is not clear. The authors decided to modify the measurement scale by using binary responses, rated from 0 to 1. However, in this paragraph, they mentioned that the cut-off point must be 2, following this reference “In a previous study in medical students, the cut-off used was 2 or more with the mean score of the whole sample was 5.77” (see p.4, line 171-172). How is it possible to use the cut-off of 2 as a threshold for the mean score based on 0/1 responses?

Line 212 ‘An univariable linear regression model was generated’ – it is important to provide a reference that allowed the application of this statistical tool for the current sample, considering its relatively small size.

Results

Line 225. In Table 1, p-values are related to what kind of statistical analysis? In this table, the mathematical calculation of pandemic participants results in 43, however, the authors mentioned 45 in the heading. Thus, an explanation is needed.

The mean examination is the central point of this article. However, the authors provided only a graph to demonstrate the differences between the pandemic and pre-pandemic outcomes. Arguably, t-tests or any other relevant statistical procedures would be more reliable instruments to catch the variations. Moreover, the description of a univariable linear regression does not follow APA style, thus, it is quite complicated to comprehend the meaningfulness of this procedure.

Line 255 – the rules for categorization of psychological distress did not describe properly.

In Discussion section, the authors mentioned the Golem effect as an explanation of findings. However, it is not clear enough how they measured it in this study. The explanation is necessary.

Reviewer 2 Report

I suggest a minor review of the current manuscript. I hope the authors can improve the manuscript.

Psychological distress and exam performance among paramedics’ students in Geneva during the COVID-19 pandemic: a mixed methods study

Introduction

The authors need to justify why psychological distress was caused by COVID-19 and the pre-entry of the paramedics' students in emergency services due to COVID-19. The issue of students' lack of skills and capacity must be narrated as the reason for this study (authors already reported that in the discussion section). Lack of practice, teaching instruction delivery and high pressure on new students are some of the major causes of students' psychological stress with the online teaching method warranting the current study.      

Methodology

The authors need to justify the six interviews for the qualitative study. How was the saturation achieved here?

Whenever authors use the abbreviation, please offer the full form and use abbreviations like (L.R) and (F.O), etc.

Why do authors conduct qualitative semi-structured interviews if themes are already identified? Qualitative research design is exploratory or confirmatory?

Results 

The results must be for 118 samples in Table 1. The total count for pre-pandemic and pandemic is (75+45=120). Please correct it. Instead of using Sex, we must use the term ‘Gender’ in Table 1. In Table 2. What are the scores provided? It is a CGPA or something else. Please offer clarity. Figure 2 is not clearly explained. Also, discuss and explain the 48.1% of psychological distress among students.

In the qualitative analysis, the authors need to offer specific comments from the respondents highlighting the issue.

Discussion

No suitable discussion is offered in the present work, and the authors must discuss study findings and compare the current study results with previously published results. A common suggestion is that the discussion must link with the research question and objectives.

Conclusion

The section must discuss the theoretical and practical implications that may help the practice or contribute to the theory in its current form; no theoretical implication is documented. The study limitation was not fully presented, leading to future research work.

General

Following the journal guidelines, the referencing style was not consistently used in the current manuscript. 

Good Luck

Reviewer 3 Report

Manuscript ID: ijerph-2206743

Dear Authors,

The topic of the paper is very interesting. The COVID-19 pandemic has forced higher education institutions around the world to implement the distance learning method in their work. This paper reveals the impact of the COVID-19 pandemic on the performance of paramedic students, and their psychological state.

To ameliorate the manuscript, some modifications may be considered.

In Section “3.3. Semi-structured Interviews” sub-headings such as “The experience of semi-lockdown”, “Stress and emotions”, etc. should also be presented in numbers (e.g. 3.3.1. The experience of semi-lockdown, 3.3.2. Stress and emotions…).

The "Discussion" section should not contain subsections. It should be written as a whole with limitations and further perspectives.

The sentence "Future research is needed..." is repeated in the Perspectives and Conclusions sections. I suggest revising this sentence in the Conclusion.

The Conclusions section should follow the research results, which means it should be more detailed.
